# Conformational dynamics in crystals reveal the molecular bases for D76N beta-2 microglobulin aggregation propensity

Tanguy Le Marchand[1], Matteo de Rosa [2], Nicola Salvi [3], Benedetta Maria Sala[2], Loren B. Andreas[1], Emeline Barbet-Massin[1], Pietro Sormanni[4], Alberto Barbiroli[5], Riccardo Porcari[6], Cristiano Sousa Mota[7], Daniele de Sanctis[7], Martino Bolognesi [2,8], Lyndon Emsley[1], Vittorio Bellotti [6], Martin Blackledge[3], Carlo Camilloni[2], Guido Pintacuda[1] & Stefano Ricagno [2]

Spontaneous aggregation of folded and soluble native proteins in vivo is still a poorly understood process. A prototypic example is the D76N mutant of beta-2 microglobulin (β2m) that displays an aggressive aggregation propensity. Here we investigate the dynamics of β2m by X-ray crystallography, solid-state NMR, and molecular dynamics simulations to unveil the effects of the D76N mutation. Taken together, our data highlight the presence of minor disordered substates in crystalline β2m. The destabilization of the outer strands of D76N β2m accounts for the increased aggregation propensity. Furthermore, the computational modeling reveals a network of interactions with residue D76 as a keystone: this model allows predicting the stability of several point mutants. Overall, our study shows how the study of intrinsic dynamics in crystallo can provide crucial answers on protein stability and aggregation propensity. The comprehensive approach here presented may well be suited for the study of other folded amyloidogenic proteins.

[1] Centre de RMN à Très Hauts Champs, Institut des Sciences Analytiques (UMR 5280 CNRS/UCB Lyon 1/ENS Lyon), Université de Lyon, 69100 Villeurbanne, France. [2] Dipartimento di Bioscienze, Università degli Studi di Milano, 20133 Milano, Italy. [3] Institut de Biologie Structurale, CNRS, CEA, UGA, 30044 Grenoble, France. [4] Department of Chemistry, University of Cambridge, Cambridge CB2 1EW, UK. [5] Dipartimento di Scienze per gli Alimenti, la Nutrizione e l'Ambiente, Università degli Studi di Milano, 20133 Milano, Italy. [6] Wolfson Drug Discovery Unit, Centre for Amyloidosis and Acute Phase Proteins, University College London, London NW3 2PF, UK. [7] ESRF - The European Synchrotron, 38043 Grenoble, France. [8] Centro di Ricerca Pediatrica Romeo ed Enrica Invernizzi, Università degli Studi di Milano, 20133 Milano, Italy. Correspondence and requests for materials should be addressed to C.C. (email: carlo.camilloni@unimi.it) or to G.P. (email: guido.pintacuda@ens-lyon.fr) or to S.R. (email: stefano.ricagno@unimi.it)

Numerous pathologies are related to the transformation of functional well-soluble proteins into amyloid fibrils. Many involve disordered peptides or proteins (e.g., Aβ, α-synuclein). More surprisingly, globular proteins (e.g., beta-2 microglobulin, superoxide dismutase, and lysozyme) can also be associated to amyloidosis diseases[1]. Yet, evolution has highly protected their native state from aggregation, and for these proteins special conditions, such as low pH or high metal ion concentration, are required for their aggregation in vitro. In contrast, specific mutations are able to dramatically increase the potential of these proteins to form fibrils. A recent case of systemic amyloidosis involves the highly unstable D76N mutation of beta-2 microglobulin (β2m)[2]. Patients carrying this mutation exhibit large amyloid deposits in internal organs, leading to progressive gastrointestinal symptoms and autonomic neuropathy. In vitro, the native fold of the D76N mutant displays a strikingly low thermodynamic stability compared to the wild type (wt) protein[2,3]. The D76N mutant is strongly amyloidogenic and, in contrast to the wt protein, aggregates efficiently and abundantly without seeding and under native conditions[4].

Common effects of point mutations are structural perturbations of the backbone, the impairment of the hydrophobic core or the introduction of hydrophobic patches on the protein surface, with potentially very drastic outcomes involving a marked change in size or in the chemical properties[1]. None of these effects can nevertheless explain the enhanced reactivity of D76N β2m. Indeed, the 76 position is located in a solvent exposed loop between β-strands E and F (Fig. 1a). The D76N mutation does not introduce a hydrophobic residue, which could lower protein solubility and Asp and Asn are isosteric. From the structural point of view the D76N mutation does not trigger major effects neither in solution nor in the crystalline state[2,4]. In particular, the crystal structures of wt β2m and of D76N mutant match very closely and the residue 76 can be well superposed in the two structures[2].

However, the crystal structure does not exhaustively define the native ensemble of a protein. Not only the position of each atom rapidly (picosecond to nanosecond) fluctuates around its equilibrium position, but also entire parts of proteins may undergo coordinated slow motions (from microseconds to longer than seconds). Such local and long-range conformational dynamics are intimately connected to biomolecular function and malfunction. Sparsely populated conformations, interconverting with the major state can be more reactive and aggregation prone, becoming a driving force in amyloid formation[1,5]. In order to better understand the amyloidogenic properties of D76N, it is therefore necessary to investigate the free energy landscape of both the wt and D76N β2m variants.

Nuclear magnetic resonance (NMR) has the unique ability to reach both spatial (atomic) and temporal (from tens of picoseconds to milliseconds) resolution. Although D76N β2m rapidly converts into non-soluble oligomers and amyloid fibrils in solution, the intrinsic dynamics of both wt and mutant forms of the protein can be studied in the crystals using solid-state NMR (ssNMR). Additionally, the absence of molecular tumbling and the capacity of the probes to achieve higher spin-lock powers in ssNMR experiments allow probing dynamics in the biologically relevant $10^{-8}$–$10^{-5}$ s timescale. The advent of fast magic-angle spinning (MAS) probes and high field magnets has recently[6,7] enabled the investigation of dynamics in samples ranging from microcrystals[8,9,11] to membrane proteins[12–14] and amyloid fibrils[15]. The wealth of crystallographic data available on β2m and the inherent protection against aggregation makes MAS NMR a technique of choice for a comparative study of the dynamics of wt and D76N β2m.

Here by combining ssNMR and ensemble modeling[16], we shed light on the determinants of the aggregation propensity of D76N

β2m. Our results show the presence of distinct minor conformers exchanging with a major form, on sub-millisecond timescales, within the crystal lattice of both wt and D76N. The minor states are characterized by a local loss of β-strand structure that is enhanced in D76N. A detailed analysis of the interaction energies of the ensembles reveals that D76 is at the center of a network of stabilizing electrostatic interactions that are lost in the D76N mutant. These observations provide a molecular explanation for the pathological aggregation propensity of D76N, together with a model that accurately predicts the impact of further mutations on β2m stability.

## Results

**Structural and biophysical studies on position 76 mutants.** Previous work suggests that D76 residue is key in determining the thermal stability and aggregation properties of β2m. Indeed, among all D to N mutations along the β2m sequence only D76N affects significantly β2m stability and aggregation propensity[17]. To strengthen this hypothesis, four mutations were inserted at position 76: D76A, D76E, D76H, and D76K. While D76K resulted too unstable to be purified, the other three variants were structurally and biophysically characterized.

The crystal structures of D76A, D76E, and D76H were determined to high resolution (Fig. 1 and Supplementary Table 1). Analogously to the structure of D76N[2], these structures do not show any major conformational change compared to wt β2m (Fig. 1 and Supplementary Table 2). However, although the backbone of the EF loop is well superimposable between the structures of wt β2m and the mutants (Supplementary Fig. 1), changing the residue in position 76 alters the H-bond network within the loop (Fig. 1), in particular in D76A and D76E. In keeping with this observation, the D76A structure where Ala76 side chain cannot establish any H-bond with neighboring residues, displays an electron density for the EF loop poorer than those of all other structures indicating an increased disorder (Fig. 1d). The last three C-terminal residues (residues 97–99) are the only stretch undergoing some rearrangement upon mutation at position 76. The side chain of Arg97 presents different conformations, and the strength of its interactions with the EF loop and to Trp95 varies in different mutants; residue Met99 interacts with Ser11 or with Arg12 (Supplementary Fig. 1). Such observations suggest a role for residue 76 in determining the interactions within the EF loop and with β2m C-terminal region.

The thermal stability and aggregation propensity of these variants were evaluated (Fig. 1g, h). Similarly to the D76N mutant, all three mutants display a decreased stability and increased aggregation propensity, confirming the key role played by position 76 in determining β2m biophysical properties. Nevertheless, despite the high resolution structural characterization of these different D76 mutants, the static crystal structures do not allow to provide a clear explanation for their distinct aggregation propensities.

**Fast protein dynamics.** Figure 2a, b show the $^{15}$N–$^{1}$H correlation spectra of wt and D76N β2m, corresponding to around 70% of the amide groups (see Supplementary Data 1 for the resonance assignments). The absent signals are associated with residues located in loops, which are subject to either large conformational disorder or extreme mobility, preventing their detection in CP-based experiments. Interestingly, signals from all residues in the BC-, EF- (encompassing position 76) and FG-loops are present in the spectra, showing that these portions of the molecule are sufficiently rigid and ordered, probably as a result of a network of strong interactions within the loops and/or with the neighboring side chains. Signals from residues 51–53 are missing

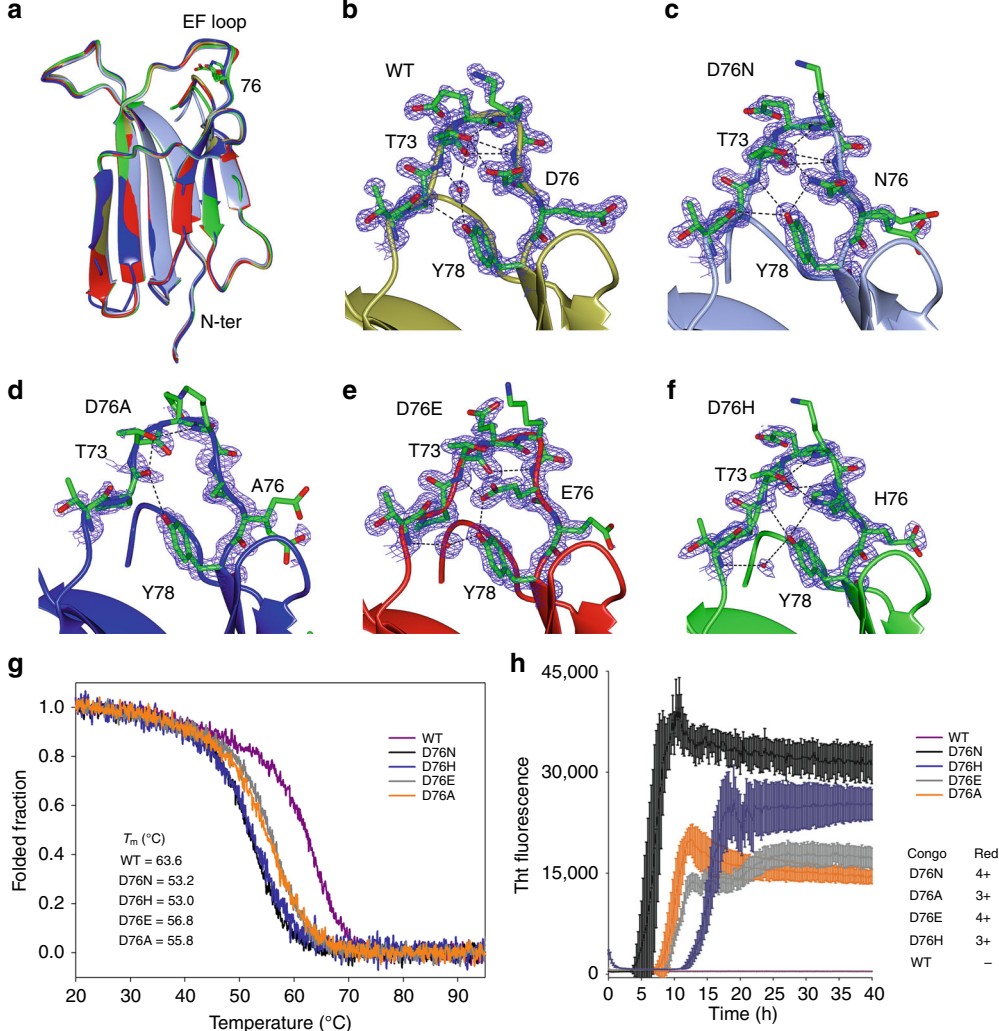

**Fig. 1** Structural and biophysical characterization of β2m mutants at position 76. **a** Superposition of the structures of wt β2m (2YXF, gold) and of the D76N (4FXL, light blue), D76A (blue), D76E (red), and D76H (green) variants. The residues in position 76 are modeled as cylinders. **b–f** Zoom-in of the EF loop of the structures shown in **a**. H-bonds are displayed as dash lines and electron density ($\sigma = 1.2$) is clipped around the EF loop. **g** Normalized temperature ramps monitored by far-UV circular dichroism at 202 nm for wt β2m and of all four 76 mutants. **h** Aggregation kinetics monitored by thioflavin T fluorescence and a Congo Red evaluation of the resulting aggregates (number are indicating the fluorescence intensity on a qualitative scale)

from the spectra, consistent with the fact that this portion of the molecule has been shown to be particularly dynamic in solution, and to display different secondary structures in different PDB structures[18].

Site-specific $^{15}N$ spin–lattice relaxation rates in the laboratory ($R_1$) and rotating frame ($R_{1\rho}$)[9] were measured for the two molecules (pulse sequences and typical decay curves are shown in Supplementary Fig. 3), and are depicted in Fig. 1c, d (see also Supplementary Data 2). These rates are mainly determined by the fluctuation of the $^{15}N$ chemical shift anisotropy (CSA) and of the $^{15}N–^1H$ dipole coupling. These data were interpreted using the simple model-free approach[19], determining order parameters ($S^2$) and correlation times ($\tau$) for all visible amide groups, which respectively report on the amplitude and on the timescale of local motions (Fig. 2e, f and Supplementary Data 3). For most of the residues, the obtained order parameters are close to 0.9, and extremely similar for the two variants. These results indicate that the two proteins are overall very rigid, as expected for a beta-sandwich fold and that the rigidity of the beta-sheet core of the protein is not affected by the mutation. The residues in the EF-loop (containing residue 76) display lower order parameters. While this is in principle not surprising, as loops may feature

significant mobility also in the presence of crystal packing, it is intriguing to observe that the fitted order parameters are consistently lower in D76N than in wt β2m, reflecting an increase in flexibility upon mutation. Small differences in the order parameters were also observed for residues 11 and 12 at the end of the A-strand. To better compare these data with crystallographic data, the crystal structures of wt and D76N were determined from diffraction data collected at room temperature (RT) rather than at 100 K. The RT structures are highly comparable in terms of conformation and B factors with structures from frozen crystals (Supplementary Fig. 2 and Supplementary Table 2).

Molecular dynamics (MD) simulation offers a mechanistic description of fast protein dynamics[20]. We compare the experimental order parameters with those calculated from two 50 ns MD runs (Fig. 2e) of both wt and D76N β2m in a water box. Overall, the simulation reproduced the experimental sequence dependence with good accuracy. In the simulation, the AB loop and the C-terminal region are predicted to be highly flexible, more so in the D76N mutant than in the wt, in agreement with the fact that these regions are not visible in CP-based experiments. The high-quality reproduction of dynamics

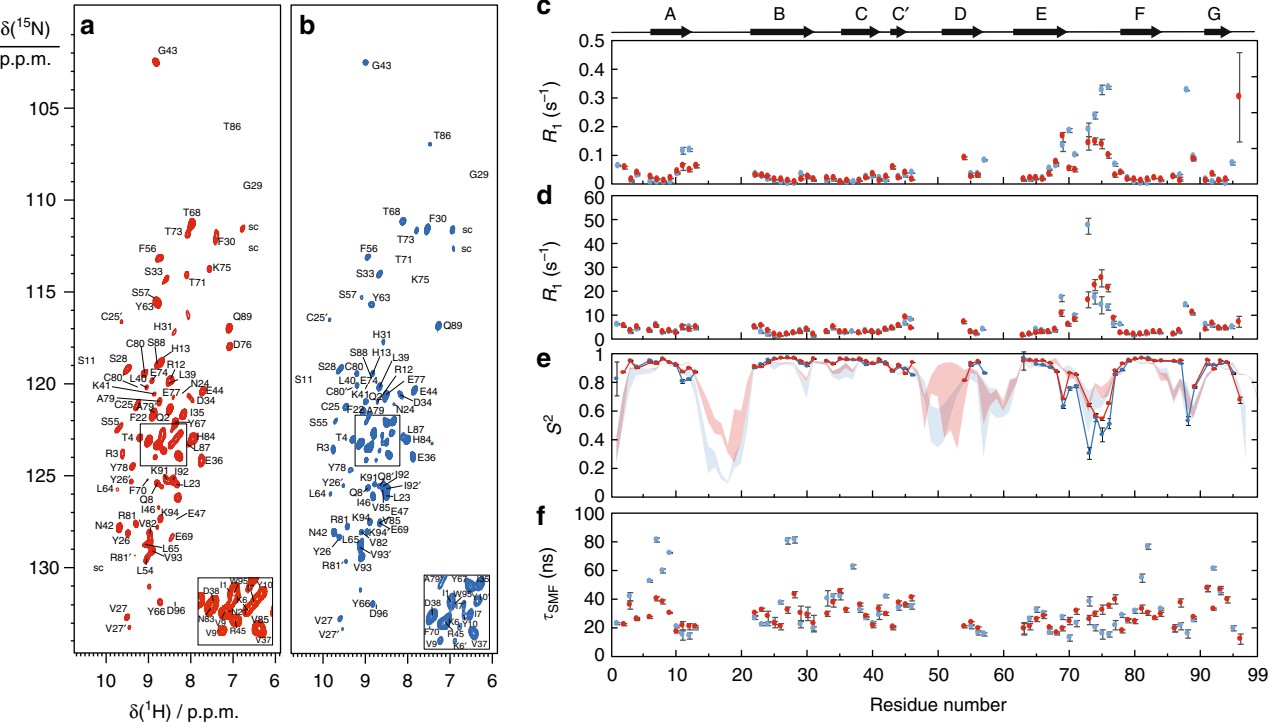

**Fig. 2** Detection of fast local motions by $^{15}$N $R_1$ and $R_{1\rho}$ relaxation experiments. $^{15}$N–$^1$H CP-based HSQC of wt- (**a**) and D76N- (**b**) [$^2$H, $^{13}$C, $^{15}$N] β-2 microglobulin with 100% reprotonation at the exchangeable sites recorded on an 800 MHz spectrometer at 60 kHz MAS rate (283 K, pH 5.5). All resonances were assigned using a set of experiments detailed in ref. [26]. The color code (red for wt and blue for D76N β2m) is used for all the figures presented here. Site-specific $^{15}$N $R_1$ (**c**) and $R_{1\rho}$ (**d**) relaxation rates of the two proteins were used to calculate the order parameters $S^2$ (**e**) and the correlation time (**f**) with the model-free approach. The results of a 50 ns MD trajectory are plotted together with the experimental results in E (red and blue bundles). β2m secondary structures are plotted on top of the panels. Experimental error bars (s.d.) were obtained from 1000 Monte Carlo simulations

observed throughout the two proteins experimentally and via simulation suggests that the observed motions are intrinsic to the protein and are essentially independent of immediate environment (solution and crystalline forms). It is noteworthy that the timescale of the motions detected here are in the tens of nanoseconds, which falls in the time range that is especially difficult to detect by spin relaxation in solution due to protein rotational diffusion. Reduced flexibility is found in the EF loop in both D76N and wt, possibly due to additional slower motions that are not captured by the 50 ns simulations.

**Slow protein dynamics**. The contribution of conformational exchange on microsecond to millisecond timescales to transverse spin relaxation can be characterized by measuring $^{15}$N $R_{1\rho}$ rates as a function of the amplitude of the spin-lock RF field in so called relaxation dispersion experiments. $R_{1\rho}$ was measured for all residues of wt and D76N β2m, using $^{15}$N spin-lock RF amplitudes ranging from 3 to 20 kHz. For several residues, mainly located within the β-sheet core, the relaxation rates are independent of the applied RF field. Interestingly, however, the rates of a non-negligible fraction of the signals feature ample dispersion profiles as the RF field is varied (Supplementary Fig. 4). Figure 3a shows selected dispersion profiles for residues located at the end of the A-strand and on the BC- and EF-loops. These data are indicative of conformational exchange in the microsecond to millisecond range[21].

Figure 3b reports the amplitude of the dispersion of the $R_{1\rho}$ rates ($\Delta R_{ex}$). For the wt, the residues showing the highest dispersion are located in the BC-loop, comprising Pro32, as well as residues located in the nearby N-terminus of the protein. These residues display a very similar dispersion also in the D76N

mutant, as it could be expected given the large distance with respect to the mutation site. However, in D76N, major changes are clearly observed in the EF-loop, as well as for the residues at the end of the A-strand. In the case of residues 75–76 in D76N, $^{15}$N $R_{1\rho}$ rates were so large for low spin-lock fields that the signals disappeared within the shortest relaxation delay, indicating extensive exchange contribution for these two residues.

These results reveal the presence of significant conformational exchange, consistent with two global processes. A first mode, common to wt and to D76N variant, involves the residues in spatial proximity to Pro32. A second mode affects the EF-loop and remote residues at the end of the A-strand, and produces a clear experimental signature only for D76N β2m.

**Replica-averaged metadynamics (RAM) ensemble simulations**. In order to gain further insights into the conformational exchange process and its consequences for the amyloid fibril formation, RAM simulations were performed[22,23]. RAM is used to integrate experimental data, in this case ssNMR chemical shifts, in molecular force fields to accurately explore the conformational space over timescales longer than those accessible during conventional MD runs.

On the basis of $^1$H$^N$, $^{15}$N, $^{13}$Cα, $^{13}$Cβ, and $^{13}$C′ chemical shifts, converged (Supplementary Figs. 5 and 6) ensembles of structures were calculated for both the wt and the D76N mutant. Free energy surfaces were generated (Fig. 4a, b) as a function of the side-chain rotamer distribution (AlphaBeta collective variable (CV), AB, see Methods) and the antiparallel β-structure content (AntiBetaRMSD CV, anti-β, see Methods). In both cases, the calculated structures clustered around two bundles, describing a major populated substate close to the crystal structure (less than

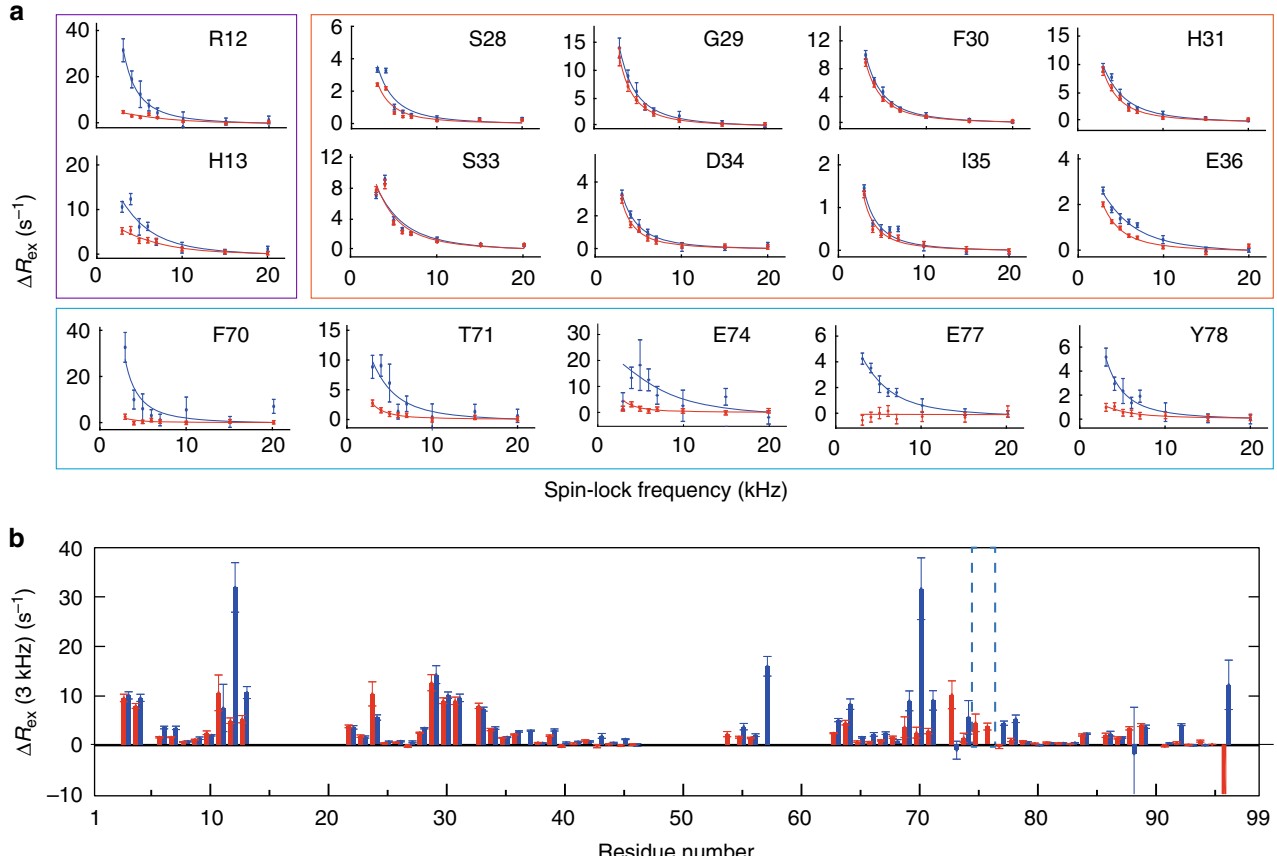

**Fig. 3** Conformational exchange of β2m probed by relaxation dispersion experiments. Selected examples of $^{15}$N $R_{1\rho}$ relaxation dispersion curves for wt (red) and D76N (blue) β2m. **a** The purple, orange, and blue rectangles show respectively residues located at the end of the A-strand, in the CD-loop and EF-loop. Best Lorentzian fits are depicted with solid lines. The difference between $^{15}$N $R_{1\rho}$ rates at 3 kHz and the fitted values at 20 kHz spin-lock field are depicted versus the sequence in **b**. Error bars (s.d.) were obtained from 1000 Monte Carlo simulations

2 Å full backbone average RMSD) and a minor populated substate, which features significant differences in secondary structural content (Fig. 4c, e) and chemical shifts differences qualitatively in agreement with those measured (Supplementary Fig. 5). The major substate (85 and 95% populated for the wt and D76N, respectively) is characterized by a high β-sheet content (~48% for both ensembles) while the minor substates are characterized by a significantly lower β-sheet content (~38 and 30% for wt and D76N, respectively).

An analysis of the predicted solubility of the two ensembles performed with CamSol[24] shows that the D76N mutation in the sequence causes a small reduction of the solubility for residues 74–77 (Supplementary Fig. 7). The solubility profiles corrected for the structure and averaged over the ensembles show more distributed differences (Fig. 4d, f, black shades), with a general trend of reduced solubility for the D76N mutation but with no significant changes. The same analysis performed instead only on the minor substates (Fig. 4d, f, light blue shades), reveals a loss of solubility for the D76N variant. In particular, the number of residues below the aggregation prone threshold increases from one to four for the D76N minor substate and all the aggregation-prone sites belong to regions (around residues 65 and 85) that have been recently shown to be crucial for modulating the aggregation properties of the protein[25]. Comparison of the secondary structure content and the solubility profiles in the minor substates shows how the larger loss of beta structure in the A-strand for the D76N variant with respect to the wt corresponds to a decrease of solubility for B-strand region that is left more exposed to the solvent.

**Interaction network analysis.** The RAM results provide a molecular framework for understanding the dramatic increase in the aggregation propensity of the D76N mutant, but do not explain why the mutation is also reducing the overall stability of the protein.

In order to understand the energetic contribution of the D76 mutation, we calculated the nonbonded interaction energies for each pair of residues, averaged over the ensemble, thus resulting in two interaction matrices. By subtracting the wt matrix from the D76N matrix, we identified the largest energy gap in the distribution of the elements. We then performed a network analysis based only on the residue pairs for which the energy difference was larger than the gap, i.e., on residue pairs experiencing large differences in their interaction energy between the two ensembles. Figure 5a shows the result of this network analysis in the form of a graph.

Not surprisingly, it is evident that the largest differences in nonbonded interaction energies between wt and D76N are related to the interactions of residue 76 with the neighboring residues (Fig. 5a). Mutation of D76 to N weakens a number of interactions with charged residues. In particular a salt-bridge with K41 is lost and the interaction with the R97 side chain that mediates an interaction between the C- and the N-termini of the protein disappears. Overall this explains the higher loss of β structure at the N- and C-termini in D76N minor substate, thus linking the lower fold stability with the increased aggregation propensity in such minor substate.

The analysis of the ensembles suggests that D76 is in a key position with respect to the electrostatic interactions among

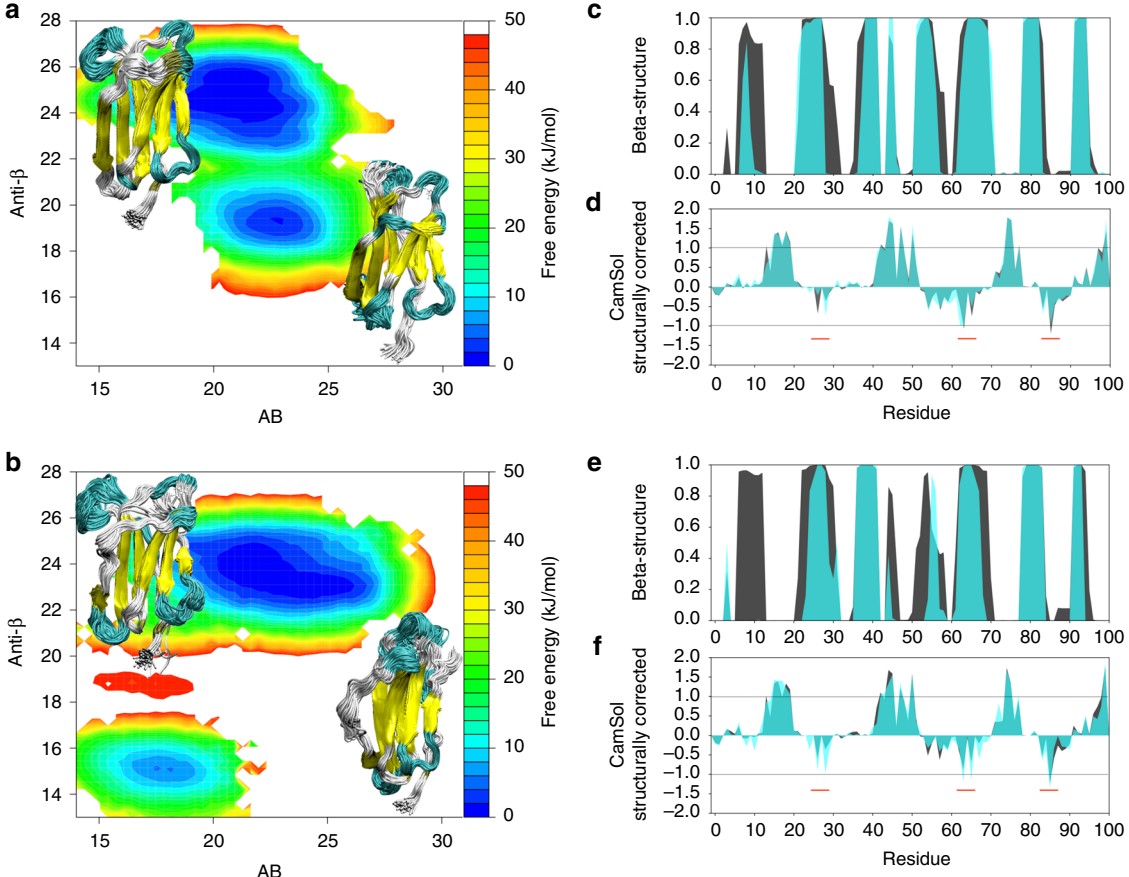

**Fig. 4** Conformational ensembles for native β2m variants. Free energies surfaces (kJ/mol), secondary structure content and solubility for the wt (upper panels) and D76N (lower panels) ensembles. **a**, **b** Free energy surfaces for the wt and D76N variant, respectively. The surfaces are shown as a function of the side chains rotameric state, AB, and the antiparallel β-structure content (anti-β) with a set of representative conformations for the major and minor substates. **c**, **e** Beta structure content per residue (within each ensemble) for the wt and D76N variants, respectively. The major state is shown in black, while the minor in cyan. **d**, **f** CamSol solubility scores for wt and D76N, respectively. Scores greater than 1 suggest a highly soluble site while a score lower than −1 suggest an aggregation-prone site. The major state is shown in black while the minor in cyan

surface residues. In order to confirm the role of the long-range interaction between R97 and D76, the R97Q mutant was prepared: in keeping with the above considerations, the mutant protein presents a significantly lower $T_m$ value and intriguingly Q97 side chain is disordered in the structure (Fig. 5b, c). Finally, to further strengthen the hypothesis about the importance of surface electrostatic interactions centered around D76 in determining the stability of β2m, the nonbonded energy was recalculated from the RAM simulation of the wt by modifying one-by-one the charge state for all the D–N mutants previously reported[17], as well as for R97Q. The resulting energies were then compared with the corresponding melting temperatures (Fig. 5d) showing a remarkable correlation. Particularly relevant is the agreement between the prediction of the increased stability for the D98N mutant and the effect of the R97 mutation: this shows an energy increase higher than any of the D–N mutations but lower than the N76 mutant in agreement with the hypothesis that the interaction between the EF loop and the C-terminal residues have an important role for the stability of the protein.

**Structural investigation of D76N β2m fibrils by ssNMR.** In order to link our observations to the mechanism of aggregation, we investigated amyloid fibrils grown in vitro by ssNMR. Resonance assignment was performed using the same $^1$H-detected correlation experiments as for the resonance assignment of the

microcrystalline form ref. [26] (Fig. 6a). Sequential resonance assignment was obtained for 53 residues, suggesting that this portion of the molecule corresponds to the most rigid and ordered core of the fibrils. The secondary structure of β2m in the fibrils was determined with TALOS-N software[27] on the basis of $^1$H$^N$, $^{13}$CA, $^{13}$CB, $^{13}$CO, and $^{15}$N chemical shifts (Fig. 6b, c). Five β-strands are predicted to be present in the fibrils. Interestingly, the native B-strand is shortened at the N-terminal side upon fibril formation. Residues corresponding to the C′-strand are not observed, indicating disorder or substantial backbone flexibility. The D-strand is extended toward the N-terminus, the E-strand is subject to an important rearrangement and the EF-loop is involved in β structure. The $^{13}$CA and $^{13}$CB chemical shifts of C25 (55.6 and 39.3 p.p.m., respectively) are indicative of an oxidized state as in the native monomer[28]. This is in agreement with previous data showing that the disulfide bond is necessary for the growth of wt β2m fibrils[29].

We next aimed at characterizing the tertiary/quaternary structure, defined as the arrangement of the intermolecular β-strands in fibrils[30]. We thus performed $^1$H–$^1$H through-space correlations[31], using radio frequency-driven recoupling mixing[32]. These experiments were acquired as 3D or 4D datasets, where the contacts between amide protons were encoded with the chemical shifts of the $^1$H and/or $^{15}$N atoms of starting and/or landing amide groups. These experiments revealed contacts between residues spaced in sequence by less than five residues. This

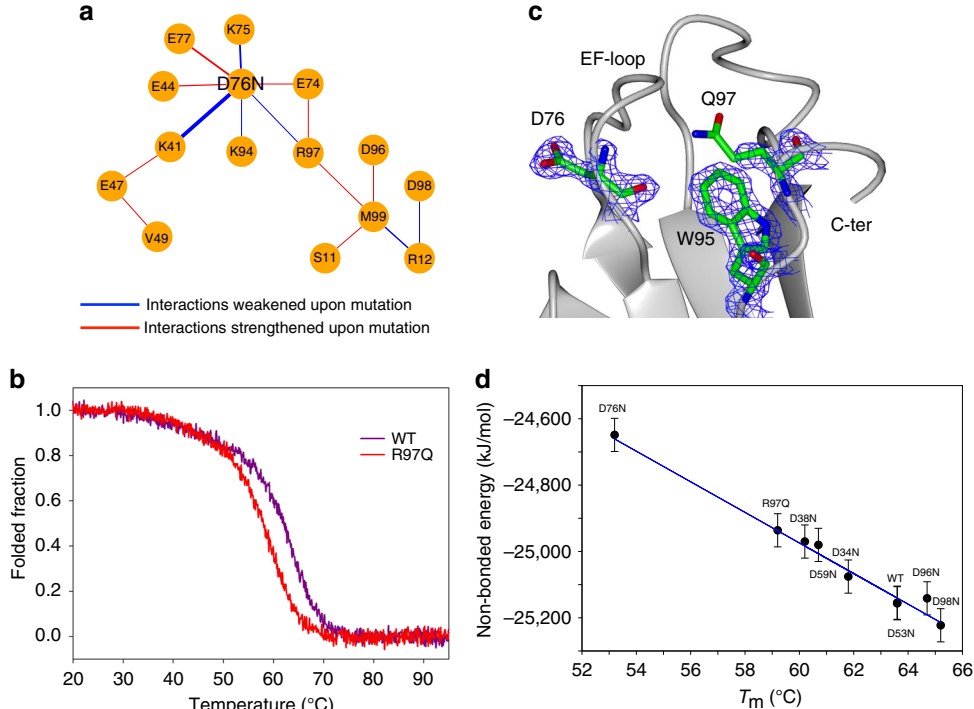

**Fig. 5** Perturbation of the energy network upon D76N mutation. **a** The network shows the interactions between couples of amino acids that are strengthened (red) or weakened (blue) upon mutation as calculated over the two ensembles of structures. **b** Temperature ramp of wt and R97Q variants monitored by far-UV ($T_m$: WT = 63.6 °C, R97Q = 59.2 °C). **c** Crystal structure of the R97Q mutant. Electron density (2FoFc) at 1.2 σ is clipped around residue D76, W95, and Q97 shown as cylinder models. The lack of electron density for Q97 side chain indicates its high flexibility. **d** Correlation between thermal stabilities for multiple β2m variants and the predicted change in energy obtained from the wild type ensembles by removing (or inverting) the side chain charges. Error bars (s.e.m.) where obtained through a block-averaging procedure

pattern is strongly indicative of a parallel, in register β-sheet intermolecular arrangement, where each residue faces its homologous residue in a neighboring molecule.

## Discussion

In 2012 the β2m D76N mutant was reported to be responsible for an aggressive form of systemic amyloidosis with large fibrillar deposits in internal organs. Exceptional D76N aggregation propensity was also verified in vitro: this variant abundantly aggregates under native conditions in the absence of seeds[2]. While in principle it is unexpected that a D–N mutation on an exposed loop should trigger such a dramatic effect, the D76N mutant also displays a dramatic decrease in thermodynamic stability[4]. In contrast, the mutation of one Asp to Asn—with the consequent loss of a negative charge—in different parts of the protein did not result in any marked effects in terms of fold stability or aggregation propensity[2,17].

A saturation site mutagenesis (D76 to E, H, and A) allowed us to show here how position 76 is indeed crucial in the β2m fold, any mutation at this position results in a notable decrease in thermal stability and a remarkable increase in aggregation propensity as observed for D76N. Intriguingly, the crystal structures suggest that D76 is an important node in a network of H-bond within the EF loop. We then used ssNMR to detect the change in dynamics upon mutation. One of the advantages of ssNMR, specific to the particular case of aggregation-prone proteins, consists in studying protein dynamics in crystallo, where the crystal lattice plays a "shielding role" preserving an overall native conformation. We are therefore able to exclusively probe intrinsic dynamics of individual protein molecules and decouple them from intermolecular interactions.

A comparison of fast (sub-microsecond) timescale dynamics measured using relaxation reveals similar motional characteristics in the wt and D76N mutant, with slightly higher levels of flexibility in the mutant. The position and amplitude of the dynamic modes observed in the crystalline lattice are well reproduced by fully solvated MD simulation, suggesting that the observed dynamics are intrinsic to the protein. Remarkably, ssNMR $^{15}$N $R_{1\rho}$ relaxation dispersion experiments in crystalline β2m[11,33–35], reveal the existence of excited states that exchange with the ground state on the micro to millisecond timescale for both the wt and the D76N variant. The D76N excited state displays more extensive exchange phenomena than wt β2m, with seven sites exhibiting markedly high relaxation dispersion effects. Such sites are almost all located in the protein region neighboring the mutation: in the EF-loop, in the E-strand and at the end of the A-strand.

In keeping with these observations, ensembles resulting from RAM simulations showed that β2m displays two distinct conformational states in both mutant and wt. As sketched in Fig. 7, while globally wt β2m excited state resembles very closely the ground state, D76N populates a conformational ensemble, which is less structured. In the excited state the D and the N-terminal A-strands lose their beta structure and the C-terminus is partially detached from protein core. As a result, the aggregation-prone core strands (B, E, and F) lose the protection of the aggregation resistant edge strands (A, C, D, and G). Remarkably all the data indicate an increased level of dynamics in D76N N-terminal region that is reminiscent of the ΔN6 β2m mutant, which is known for a strong aggregation propensity[36,37]. While our data clarify the differences in aggregation propensity between D76N and wt, which likely yield to different aggregation behaviors both in vitro and in vivo,

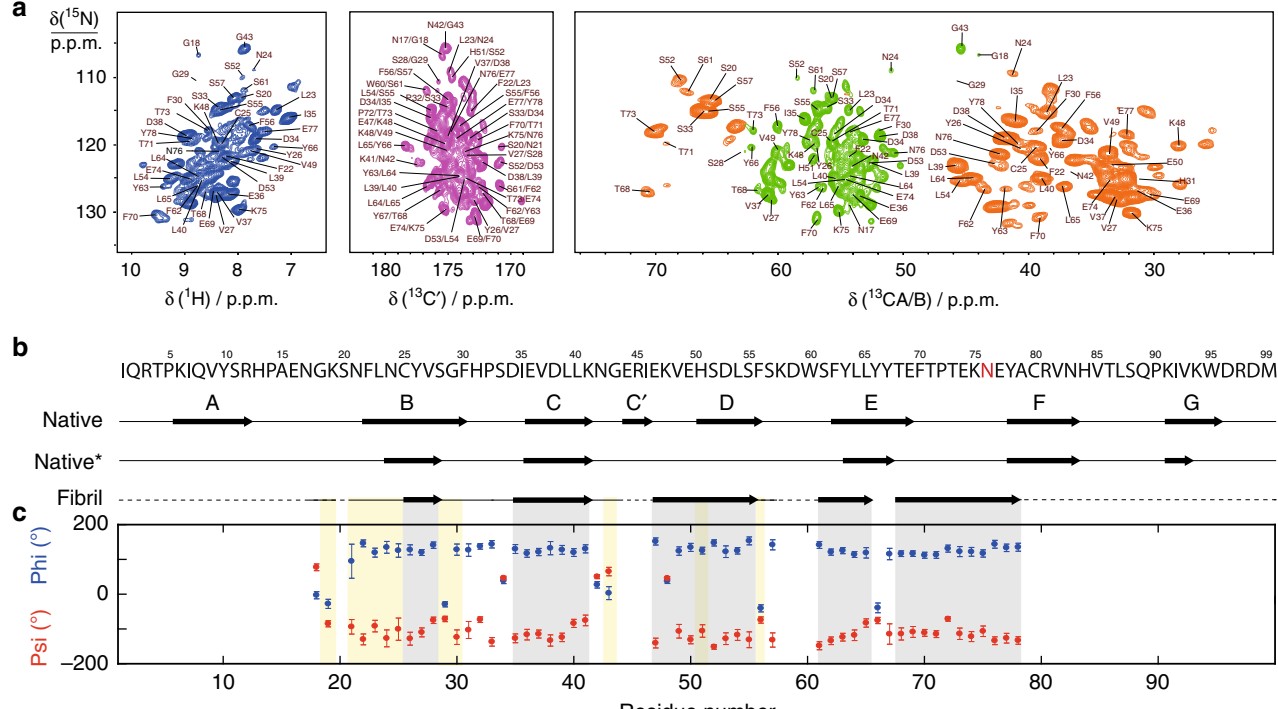

**Fig. 6** Structural characterization of D76N fibrils. **a** Assigned 1H-15N 2D projection of (H)CANH (dark blue) correlation spectrum and 13C-15N 2D projections of (H)CONH (purple), (HCA)CB(CA)NH (orange) and (H)CANH (green) correlation spectra of [2H, 13C, 15N] D76N β2m fibrils with 100% reprotonation at the exchangeable sites (283 K, pH 7). Spectra were recorded on a 1 GHz spectrometer at 60 kHz MAS rate. **b** Secondary structure prediction of D76N β2m fibrils by TALOS-N 39 (black arrows correspond to β-strands, straight lines to loops and dashed lines to an absence of prediction). The β-strands of the native fold are depicted above the primary sequence. The secondary structures of fibrillar wt β2m[40] are also shown for comparison. **c** Backbone dihedral angles phi and psi predicted by TALOS-N. The gray and yellow boxes show TALOS-N for β-strands and ambiguous predictions, respectively. Error bars (s.d.) are based on the 10 best TALOS-N database matches

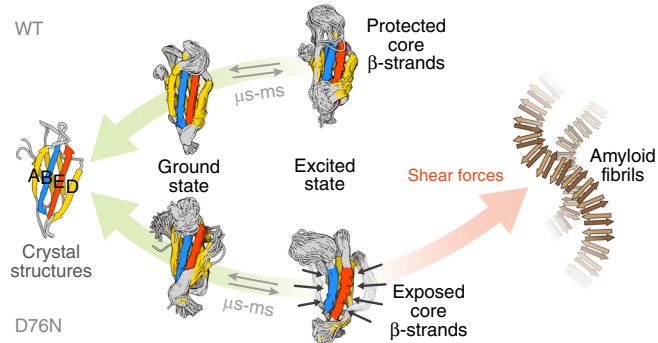

**Fig. 7** Proposed mechanism for the first steps of D76N aggregation. Under native conditions both wt and D76N are in equilibrium between low and high-energy conformations. While wt high-energy conformation has a contained aggregation propensity, the D76N excited state is less structured, highly dynamic, and exposes highly aggregation-prone regions shown in orange. Thus, modest energies such as shear forces present in the circulation system are sufficient to further unfold D76N β2m termini (model) destabilizing completely β2m fold and leading to amyloid formation

however, they do not provide specific insights on the aggregation pathways for the two β2m variants.

Furthermore, the conformational ensembles suggest a link between the differences in the excited states and the differences in the thermal stabilities where the mutation in position 76 breaks a large network of electrostatic interactions distributed over a large number of residues including the C- and the N-termini. RAM simulations indicate that the hydrophobic core around the

evolutionary conserved W95 is not optimally shielded from the solvent. A specifically designed R97Q mutant next to the C-terminus allowed us to further test this hypothesis showing that the weakening of the interaction between positions 76 and 97 significantly decreases β2m thermal stability.

Moving perspective, we complemented the analysis of the native state of D76N β2m with ssNMR characterization of amyloid fibrils grown in vitro. Interestingly, as shown in Fig. 6b, we find a progressive evolution of structural elements between the starting, native conformation derived from the crystal structure, the ensemble representing the excited state derived by RAM simulations, and the final fibrillar state. Notably, the A and G edge strands of β2m, which are destabilized by the point mutation, are not part of the cross β-structure. Additionally, the N-terminal portion of the B-strand loses secondary structure both in the excited state (native*) and in the fibrils. While these data do not represent an evidence for a particular aggregation pathway, we note that the rearrangement of the secondary structure elements in the core of the fibrils may be facilitated by the loss of structure of the corresponding native β-elements. This is particularly evident for the region encompassing residues 44–56 where the C′ and D strands are absent in the excited state while they appear to form a unique strand in the fibrillar state. A similar observation can be made for the E-strand and EF-loop region. The loss of interaction between EF-and C′D-loops, as evidenced by the network analysis, may therefore be related to their reorganization in β-structures within the fibrils.

The secondary structure of wt β2m fibrils has previously been characterized by 13C-detected ssNMR[38–40]. We note common structural and dynamical features in both wt and D76N fibrils, namely the position of the rigid core, a parallel in register beta

arrangement, the preservation of the disulfide bridge and flexible termini, but also differences can be observed in the fine details of the secondary structures, namely the positions of the β-strands. This may indicate comparable fibril architecture even in the presence of likely different aggregation pathways.

In summary, in the present work, by using a combination of X-ray crystallography, ssNMR and MD simulations, we have been able to determine the impact of the highly pathogenic D76N mutation on the native state of β2m. Thermodynamic stability of the native fold is lowered due to the weakening of a set of electrostatic interactions and to the diminished interaction between the C-terminal residues and the β2m hydrophobic core. Enhanced aggregation propensity in D76N appears to be correlated with the existence of a high-energy state characterized by a loss of structure in the protective edge β-strands. Previous observations that low energetic contributions, e.g., the friction in extracellular space in vivo or the shear forces at the interface air/liquid in vitro are sufficient to trigger amyloid aggregation[2,4], are well rationalized by the presence of an amyloid-competent state under native conditions. In conclusion, our observations provide a molecular explanation for the biophysical properties leading to D76N pathogenicity, while showing the effectiveness of our integrated approach to investigate the molecular bases of amyloid aggregation propensity, a method that may well be applied to other amyloidogenic folded proteins.

## Methods

**Protein production.** β2m mutants (D76E, D76A, D76H, D76K, and R97Q) were produced using the Phusion Site-Directed Mutagenesis Kit (lifetechnologies), following the manufacturer protocol. Forward primers used for mutagenesis are available in Supplementary Table 3. Any attempt to obtain amount of the D76K mutant suitable for structural and biophysical characterization failed. All other variants together with wt and D76N proteins were instead expressed and purified using standard protocols[36].

**Crystallization structure determination and analysis.** All the proteins were crystallized under the standard conditions as used for most of the β2m monomeric variants: protein concentration 8–10 mg/ml, 21–27% w/v PEG 4000, 15% v/v glycerol, 0.2 M ammonium acetate, 0.1 M sodium acetate, pH 5.0–5.5.

Crystals were flash-frozen to 100 K in liquid nitrogen without supplementation of additional cryoprotectants. Diffraction data were collected at the European Synchrotron Radiation Facility (Grenoble France) at the ID29, ID23-2, and ID23-1 beamlines[41–43].

Data collection at RT was performed as follows. Beta2-microglobulin crystals were mounted on a microRT Room Temperature kit (MiTeGen), sealed with a MicroRT polyester capillary, filled at the extremity with 5 μl mother liquor. RT X-ray diffraction data were collected at beamline ID29 at ESRF using Pilatus3 6 M detector in a shutterless mode at 0.9840 Å wavelength. Data collection strategy were calculated using BEST[44] from two images collected with a 50 × 30 μm² Gaussian beam and a flux limited to 5.58 × 10¹⁰ photons/s (corresponding to 5% of the full beam) to limit radiation damage. Strategy was calculated using a susceptibility factor of 40 to take into account the faster decay rate that occurs at RT. For the wt crystal, 880 images were collected with 0.02 s exposure time and 0.15° oscillation angle, with a flux of 1.12 × 10¹⁰ photons/s (corresponding to 1.35% of full beam). For the D76N mutant, 1800 images were collected with 0.02 s exposure time and 0.1° oscillation angle, with a flux of 8.30 × 10⁹ photons/s (1% of full beam).

Data were processed with MOSFLM[45] and either SCALA or AIMLESS[46] and structures solved by molecular replacement using the program PHASER[47] and wt β2m as search model (pdb ID 2YXF). All the structures were refined with Phenix refine[48] and manual model building and structure analysis were performed with COOT[49]. CCP4MG was used for the preparation of the figures[50].

**Thermal denaturation.** Thermal unfolding was monitored at 202 nm in the 20–95 °C range using a JASCO J-810 spectropolarimeter equipped with a Peltier device and a fluorescence detector (JASCO corporation, Tokyo, Japan). For all β2m variants involved in this study ramps were carried out in 50 mM sodium phosphate, pH 7.4. Protein concentration was set to 0.1 mg/mL (cell path 0.1 cm), and the temperature increase was 50 °C/h. $T_m$ values were determined as the minima of the first derivative of the unfolding profiles. Thermal unfolding experiments display high reproducibility.

**Aggregation propensity.** A volume of 100 μL of recombinant variants D76N, D76A, D76H, D76E, and wt β2m at 40 μM (PBS pH 7.4) containing 10 μM Thioflavin T (ThT) (SIGMA), were incubated at 37 °C in Costar 96-well black-wall plates closed with clear sealing film (4TITUDE) and were subjected to 900 rpm double-orbital shaking. ThT fluorescence was recorded every 15-min (BMG LABTECH FLUOstar Omega). Experiments have been performed in triplicate.

**Isotopically labeled sample preparation.** Isotopically labeled D76N and wt β2m were expressed in triple labeled (²H, ¹³C, ¹⁵N) M9 medium (D2O, d₇-¹³C-glucose, and ¹⁵N-ammonium chloride). The purification and refolding step were performed as described above in non-deuterated water allowing proton back-exchange at every amide site. Crystals were grown using the sitting drop vapor diffusion technique (10 mg/mL protein concentration, crystallization solution: 22% w/v PEG, 20% v/v glycerol, 0.1 mol/L sodium acetate, pH 5.5). After 2 weeks, microcrystals of D76N and wt β2m were filled into 1.3 mm NMR rotor by centrifugation (14,000 r. p.m.).

**NMR spectroscopy.** NMR experiments were conducted on 18.8 or 23.5 T narrow-bore Bruker Avance III spectrometers, corresponding respectively to ¹H Larmor frequency of 800 MHz and 1 GHz at 60 kHz MAS. Sample temperature was monitored by bulk-water chemical shift using the PEG line as a reference and set to 283 K.

Resonance assignment of D76N β2m was performed with the same procedure that described[51] using the whole set of experiments ((H)CANH, (H)(CO)CA(CO)NH, (H)CONH, (H)CO(CA)NH, (H)(CA)CB(CA)NH, and (H)(CA)CB-(CA)(CO)NH).

The pulse sequences used for ¹⁵N $R_1$ and $R_{1\rho}$ relaxation rates measurement are as described[9] (Supplementary Fig. 1). Relaxation delays were set to 0, 0.5, 1, 2.5, 5, 10, 20, and 40 s for ¹⁵N $R_1$ measurements and 1, 3, 8, 20, 45, 100, 180, and 300 ms for ¹⁵N $R_1$rho measurements. The ¹⁵N $R_{1\rho}$ relaxation dispersion curves were obtained with ¹⁵N spin-lock frequency of 3, 4, 5, 6, 7, 10, 15, and 20 kHz. Examples of decay curves are shown in Supplementary Fig. 3.

The NMR data processing was done with Topspin, the resonance assignment with CARA and the analysis of the relaxation with Sparky.

**Relaxation data fitting.** All relaxation data fitting has been done using Matlab. Amide ¹⁵N relaxation was assumed to be dominated by the dipolar coupling with its bonded ¹H and by its CSA. The relaxation rates can therefore be expressed as:

$$R_1 = \frac{d^2}{4}[J_0(\omega_H - \omega_N) + 3J_1(\omega_N) + 6J_2(\omega_H + \omega_N)] + \frac{\omega_N^2 \Delta\sigma^2}{3} J_1(\omega_N), \quad (1)$$

$$R_{1\rho} = \frac{d^2}{8}[4J_0(\omega_{SL}) + J_0(\omega_H - \omega_N) + 3J_1(\omega_N) + 6J_1(\omega_H) + 6J_2(\omega_H + \omega_N)]$$
$$+ \frac{\omega_N^2 \Delta\sigma^2}{18}[4J_0(\omega_{SL}) + 3J_1(\omega_N)], \quad (2)$$

where $d = \frac{\mu_0 \hbar \gamma_N \gamma_H}{4\pi r_{NH}^3}$, $\mu_0$, the vacuum permittivity, $\gamma_H$ and $\gamma_N$, respectively, the ¹H and ¹⁵N gyromagnetic ratios, $r_{NH}$ the NH bond distance (1.02 Å), $\omega_H$ and $\omega_N$, respectively, the ¹H and ¹⁵N Larmor-precession frequency, $\Delta\sigma$ the ¹⁵N CSA (170 p. p.m.), and $\omega_{SL}$ the rf spin-lock frequency.

We applied the Lipari–Szabo model-free analysis. In this framework, the spectral density function $J_m(\omega)$ is independent of the value of $m$ and is given by:

$$J_{SMF}(\omega) = \frac{2}{5}\left[\frac{(1 - S^2)\tau_{eff}}{1 + (\omega\tau_{eff})^2}\right], \quad (3)$$

where $S^2$ is the order parameter and $\tau_{eff}$ the effective model-free timescale. This approach does not account for the anisotropy. Intensity decay was therefore treated as a mono-exponential process.

The relaxation decay curves were fitted by minimizing the chi-square function:

$$\chi^2 = \frac{1}{N}\sum_{k=1}^{N}\frac{\left(I_{X,exp}(t_k) - I_{X,calc}(t_k)\right)^2}{\sigma_{X,exp}^2}, \quad (4)$$

$X$ stands for ¹⁵N $R_1$ or ¹⁵N $R_{1\rho}$, $N$ is the number of points of the curve, and $\sigma$ is the experimental error calculated as the spectrum RMSD.

Error estimate was done with 1000 Monte Carlo simulations. In these simulations, synthetic datasets are produced by adding Gaussian random noise to the back-calculated decay curves. The error was then defined as the standard deviation of the ensemble of dynamical parameters obtained by fitting the synthetic datasets.

**MD simulations.** The simulation system consists of either wt or D76N β2m in a dodecahedron box. Na⁺ and Cl⁻ ions were added to neutralize charges. Two 50-ns independent simulations were seeded from both the wt (PDB code 2YXF) and mutant (4FXL) crystal structures. GROMACS[52] was used to calculate the

trajectories. Prior to the production runs of 50 ns with an integration step of 2 fs in the NPT ensemble, energy minimization with the steepest-descent minimization algorithm is performed, followed by 2-ns equilibration steps in the NPT ensemble. Amber ff99SB-ILDN[53] is used as force field with TIP4P/2005[54] water model. Holonomic constraints are applied on all hydrogen-heavy atom bond terms to remove fast modes of oscillation. A cutoff of 10 Å is used for the Lennard-Jones and electrostatic interactions. Particle-mesh Ewald summation with a grid spacing of 16 Å is used to calculate long-range electrostatic interactions. A modified Berendsen thermostat with a damping constant of 0.1 ps is used to keep the temperature of the system at 298 K. A Berendsen barostat with a relaxation time of 2 ps is used to control the pressure at the target value of 1.01325 bar.

**RAM simulations.** The simulations were carried out using GROMACS[52] and PLUMED[55]. The system was described using the Amber03W[56] force field in explicit TIP4P05 water at 278 K. The starting conformations were taken from the 2YXF and 4FXL X-ray structures for the wt and D76N mutant, respectively. The structures were protonated and solvated with ~8200 water molecules in a dodecahedron box of ~260 nm$^3$ of volume. The RAM protocol was applied using chemical shifts as replica-averaged restraints[54] and bias-exchange metadynamics[23,57,58]. Four replicas of the system were simulated in parallel with a restraint applied on the average value of the back-calculated NMR chemical shifts with a force constant of 24 kJ/(mol p.p.m.$^2$).

Each of the four replica is biased along one of the following four CVs: the antiparallel β content (the "anti-β" CV), the parallel β-sheet content (the "Para-β" CV), the AlphaBeta CV defined over all the chi-1 angles for the hydrophobic side-chains (the "AB" CV), and the AlphaBeta CV defined over all the phi and psi backbone dihedral angles of the protein (the "bbAB" CV). Definition of the CVs are available in the PLUMED manual and publication. Gaussians deposition was performed with σ values set to 0.1, 0.04, 0.16, and 0.25, for β, Pβ, AB, and bbAB, respectively; an initial energy deposition rate of 0.125 kJ/mol/ps and a bias-factor of 8. Each replica has been run for a nominal time of 700 ns, with exchange trials every 50 ps.

The cumulative sampling of the four replicas was used to generate a free energy landscape as a function of the before mentioned CVs. A set of microstates is identified by dividing the four-dimensional CV-space into a homogeneous grid of small dimensional hypercubes and their free energy is obtained using a standard weighted histogram analysis[59]. Lower-dimensional FESes are obtained integrating out the CVs not showed.

**Data availability.** Coordinates and structure factors for D76E, D76A, D76H, R97Q, wt RT, and D76N RT have been deposited in the Protein Data Bank under accession codes 4RMU, 4RMW, 4RMV, 5CSG, 5CS7, and 5CSB, respectively. The datasets generated and/or analyzed during the current study are available from the corresponding authors upon reasonable request.

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

## Acknowledgements

C.C. acknowledges the CINECA award under the ISCRA initiative, for the availability of high performance computing resources and support. We thank the members of the technical staff of the Institut des Sciences Analytiques for assistance with the NMR spectrometers. Financial support from the CNRS (IR-RMN FR3050) and from the European Research Council (ERC) under the European Union's Horizon 2020 research and innovation program (ERC-2015-CoG GA no. 648974) are gratefully acknowledged. L.B.A. was supported by a MC incoming fellowship (REA Grant agreement no. 624918 "MEM-MAS"). S.R. was funded by the Italian Ministry of University (Project FIRB RBFR109EOS) and by Fondazione Cariplo (Cariplo Giovani Project 2016-0489).

## Author contributions

T.L.M., M.D.R., N.S., B.M.S., L.B.A., P.S., E.B.-M., A.B., R.P., C.S.M., D.D.S., and C.C. performed the experiments; M.Bol., L.E., V.B., M.Bla., C.C., G.P., and S.R. designed the study; T.L.M., C.C., G.P., and S.R. wrote the paper with contributions from all other authors.

## Additional information

**Competing interests:** The authors declare no competing interests.

