## [Peer Review File · Nature Communications]

This manuscript has been previously reviewed at another journal that is not operating a transparent peer review scheme. This document only contains reviewer comments and rebuttal letters for versions considered at Nature Communications. Mentions of prior referee reports have been redacted.

REVIEWERS' COMMENTS:

Reviewer #1 (Remarks to the Author):

The manuscript by Marchand et al. describes the properties of a mutant form of b2-microglobulin (D76N) that is much more prone to amyloidogenesis than the native protein. To this end, the authors use X-ray crystallography to determine the structures of several point mutants and then analyze the fast and slow dynamics in the crystals by solid-state NMR. The experimental data suggest that D76N experiences enhanced conformational exchange and in particular destabilizes interactions within the EF loop of the protein. The experimental work is also supported by simulations and interaction analysis that propose the existence of a minor population of conformers that contain much less b-sheet character compared to the major more native conformation. In addition, the authors present solid-state NMR analysis of the secondary structure of the amyloid fibrils formed by the D76N mutant protein.

The strength of this manuscript is the use of solid-state NMR to characterize dynamics in protein crystals. While there are other examples of this in the literature, the use of this methodology here is particularly relevant as the static crystal structures do not reveal significant structural differences between the mutant forms and hence do not explain the enhanced propensity for D76N amyloidogenesis. In addition, the formation of aggregates in solution would have precluded similar analysis by solution NMR. Thus, the methodology should be relevant to other protein systems exhibiting dynamic behavior that cannot be captured by solution NMR, crystallography or cryo-EM.

The experiments presented in the paper describe the starting point of D76N amyloidogenesis and to some extent characterize the ending point (the amyloid fibril secondary structure). While the authors emphasize the role of this mutation in destabilizing the native fold of the protein, i.e. the first step on the pathway to amyloidogenesis, it would be intriguing to see in future experiments if this mutation also affects the subsequent steps in oligomerization. For example, is this mutated sequence region more prone to the formation of amyloid-type interactions? Would this explain the enhanced aggregation propensity of this mutant in comparison to other amino acid substitutions at this position? While much work will be needed in the future to fill in the gap, the experiments and conclusions in the manuscript raise intriguing questions that should be of general interest to the amyloid community.

Reviewer #1 (Remarks to the Author):

The manuscript by Marchand et al. describes the properties of a mutant form of b2-microglobulin (D76N) that is much more prone to amyloidogenesis than the native protein. To this end, the authors use X-ray crystallography to determine the structures of several point mutants and then analyze the fast and slow dynamics in the crystals by solid-state NMR. The experimental data suggest that D76N experiences enhanced conformational exchange and in particular destabilizes interactions within the EF loop of the protein. The experimental work is also supported by simulations and interaction analysis that propose the existence of a minor population of conformers that contain much less b-sheet character compared to the major more native conformation. In addition, the authors present solid-state NMR analysis of the secondary structure of the amyloid fibrils formed by the D76N mutant protein.

The strength of this manuscript is the use of solid-state NMR to characterize dynamics in protein crystals. While there are other examples of this in the literature, the use of this methodology here is particularly relevant as the static crystal structures do not reveal significant structural differences between the mutant forms and hence do not explain the enhanced propensity for D76N amyloidogenesis. In addition, the formation of aggregates in solution would have precluded similar analysis by solution NMR. Thus, the methodology should be relevant to other protein systems exhibiting dynamic behavior that cannot be captured by solution NMR, crystallography or cryo-EM.

The experiments presented in the paper describe the starting point of D76N amyloidogenesis and to some extent characterize the ending point (the amyloid fibril secondary structure). While the authors emphasize the role of this mutation in destabilizing the native fold of the protein, i.e. the first step on the pathway to amyloidogenesis, it would be intriguing to see in future experiments if this mutation also affects the subsequent steps in oligomerization. For example, is this mutated sequence region more prone to the formation of amyloid-type interactions? Would this explain the enhanced aggregation propensity of this mutant in comparison to other amino acid substitutions at this position? While much work will be needed in the future to fill in the gap, the experiments and conclusions in the manuscript raise intriguing questions that should be of general interest to the amyloid community.

Referee 1 does not suggest any amendments or modifications of the manuscript. We thank Referee 1 for the positive word that he/she uses to describe our work. We share the Referee 1 views about the direction of the future research on D76N mutant. The questions he/she raises are crucial to the full understanding of β 2m aggregation propensity.